# Impact of Interobserver Variability in Manual Segmentation of Non-Small Cell Lung Cancer (NSCLC) Applying Low-Rank Radiomic Representation on Computed Tomography

**DOI:** 10.3390/cancers13235985

**Published:** 2021-11-28

**Authors:** Michelle Hershman, Bardia Yousefi, Lacey Serletti, Maya Galperin-Aizenberg, Leonid Roshkovan, José Marcio Luna, Jeffrey C. Thompson, Charu Aggarwal, Erica L. Carpenter, Despina Kontos, Sharyn I. Katz

**Affiliations:** 1Department of Radiology, University of Pennsylvania, Philadelphia, PA 19104, USA; bardia.yousefi@gmail.com (B.Y.); maya.galperin-aizenberg@pennmedicine.upenn.edu (M.G.-A.); Leonid.roshkovan@pennmedicine.upenn.edu (L.R.); jose.luna@pennmedicine.upenn.edu (J.M.L.); despina.kontos@pennmedicine.upenn.edu (D.K.); 2Center for Biomedical Image Computing and Analytics, University of Pennsylvania, Philadelphia, PA 19104, USA; 3Perelman School of Medicine, University of Pennsylvania, Philadelphia, PA 19104, USA; lacey.serletti@pennmedicine.upenn.edu; 4Section of Interventional Pulmonology, Department of Medicine, University of Pennsylvania, Philadelphia, PA 19104, USA; Jeffrey.thompson@pennmedicine.upenn.edu; 5Division of Hematology and Oncology, Department of Medicine, University of Pennsylvania, Philadelphia, PA 19104, USA; charu.aggarwal@pennmedicine.upenn.edu (C.A.); erical@upenn.edu (E.L.C.)

**Keywords:** radiomics, interobserver variability, non-small cell lung cancer, computed tomography (CT)

## Abstract

**Simple Summary:**

Discovery of predictive and prognostic radiomic features in cancer is currently of great interest to the radiologic and oncologic community. Tumor phenotypic and prognostic information can be obtained by extracting features on tumor segmentations, and it is typically imaging analysts, physician trainees, and attending physicians who provide these labeled datasets for analysis. The potential impact of level and type of specialty training on interobserver variability in manual segmentation of NSCLC was examined. Although there was some variability in segmentation between readers, the subsequently extracted radiomic features were overall well correlated. High fidelity radiomic feature extraction relies on accurate feature extraction from imaging that produce robust prognostic and predictive radiomic NSCLC biomarkers. This study concludes that this goal can be obtained using segmenters of different levels of training and clinical experience.

**Abstract:**

This study tackles interobserver variability with respect to specialty training in manual segmentation of non-small cell lung cancer (NSCLC). Four readers included for segmentation are: a data scientist (BY), a medical student (LS), a radiology trainee (MH), and a specialty-trained radiologist (SK) for a total of 293 patients from two publicly available databases. Sørensen–Dice (SD) coefficients and low rank Pearson correlation coefficients (CC) of 429 radiomics were calculated to assess interobserver variability. Cox proportional hazard (CPH) models and Kaplan-Meier (KM) curves of overall survival (OS) prediction for each dataset were also generated. SD and CC for segmentations demonstrated high similarities, yielding, SD: 0.79 and CC: 0.92 (BY-SK), SD: 0.81 and CC: 0.83 (LS-SK), and SD: 0.84 and CC: 0.91 (MH-SK) in average for both databases, respectively. OS through the maximal CPH model for the two datasets yielded c-statistics of 0.7 (95% CI) and 0.69 (95% CI), while adding radiomic and clinical variables (sex, stage/morphological status, and histology) together. KM curves also showed significant discrimination between high- and low-risk patients (*p*-value < 0.005). This supports that readers’ level of training and clinical experience may not significantly influence the ability to extract accurate radiomic features for NSCLC on CT. This potentially allows flexibility in the training required to produce robust prognostic imaging biomarkers for potential clinical translation.

## 1. Introduction

Lung cancer is the leading cause of cancer-related death in the United States [1]. Non-small cell lung cancer (NSCLC) represents the majority of primary lung cancers and carries a poor prognosis and low overall survival [2]. Computed tomography (CT) is a routinely used diagnostic imaging tool in clinical management in oncology due to the ability of CT to noninvasively provide anatomic information for detection, staging, and therapy response assessment. Over the past decade it has become evident that quantitative features are embedded in conventional medical imaging data, not appreciable to the human eye [3]. These radiomics features are a reflection of tissue architecture, heterogeneity, and pericellular environment and can be harnessed to construct tissue signatures that correlate with clinically relevant biomarkers, including tumor histologic subtype, mutational status, degree of infiltration with tumor infiltrating lymphocytes, as well as therapeutic endpoints such as overall survival [4,5,6,7,8,9]. These imaging “phenotypes” provide valuable data that may enhance personalization of medical care in oncology [10].

It is well known that repeatability and reproducibility of radiomic features on CT are sensitive to various image details such as image acquisition settings, processing, reconstruction algorithm, and specific software used for radiomic feature extraction [5,7,9,11,12,13,14,15,16,17]. Furthermore, certain radiomic features are more sensitive to these variations than others, with first order features, specifically entropy, consistently reported as being very stable while other texture features, such as coarseness and contrast, being the least reproducible [18].

Discovery of predictive and prognostic radiomic features in cancer is currently of great interest to the radiologic community; however, there is no reliable fully automated means of segmenting lung cancer. Tumor delineation and contouring are often performed by scientists with a range of training in anatomical imaging including imaging analysts, students, physician trainees, and attending physicians using either manual or semi-automated techniques. In addition to being time consuming, 3-dimensional manual and semi-automated contouring are subject to interobserver variability. This variability has been shown to be particularly challenging with segmented lesions when associated with ground glass components and postobstructive atelectasis [4]. In order to generate high fidelity phenotypic radiomic signatures, tumor segmentations must be reproducible across different readers [17]. Performing quality segmentations is an important task. Although the ability to anticipate tumor histology, mutational status, and therapeutic consequences are all ultimate goals of radiomics, interobserver variability between readers should be thoroughly investigated before subsequent feature analysis is tested, given that these segmentations form the basis of the analyses.

To our knowledge, no study has examined how both the level and type of specialty training in manual or semi-automated segmentations affects the subsequent extraction of radiomic features. Thus, our purpose in this study is to examine how the level of specialty training impacts interobserver variability in manual segmentation and radiomic feature extraction of NSCLC on CT.

## 2. Materials and Methods

The proposed approach presents a comparative assessment of interobserver variability in segmenting NSCLC tumors on chest CTs and its effect on subsequent extraction of radiomic features and survival analysis (see Figure 1 for study schema).

### 2.1. Patient Population and Study Data

This was a single-center study with segmentations performed at our institution between July 2018 and December 2019. The CT images included in this study had slice thicknesses between 1 and 5 mm, and both contrast and non-contrast enhanced studies were included. No pre-processing methods of the CT images were employed. Two publicly available datasets containing CT images from patients with NSCLC were analyzed. The NSCLC- Radiomics-Genomics-Lung3 (also known as Harvard) dataset (Table 1) [11,19,20] contains pre-treatment CT images from 89 patients with NSCLC and the NSCLC-Radiogenomics (also known as Stanford) dataset [20,21,22] contains pre-treatment CT images from 211 patients with NSCLC and both are publicly available from the National Institutes of Health (NIH) mentioned in The Cancer Imaging Archive (TCIA) [20,21,22,23]. Patients without available imaging in the online dataset were excluded.

### 2.2. Radiomic Feature Extraction and Statistical Analysis

Radiomic features can be divided into categories, for example: first-order features, which include tissue density, shape features (i.e., volume and surface area) and texture features, describing spatial patterns of voxel intensities [5,7,9,11,12,13,14,15,16,17]. The proposed approach employs 429 radiomics features in nine categories: first-order statistics (FO) (18 features), shape-based expression (SB) (13 features), gray level co-occurrence matrix (GLCM) (23 features), gray level dependence matrix (GLDM) (14 features), gray level run length matrix (GLRLM) (16 features), gray level size zone matrix (GLSZM) (16 features), neighboring gray tone difference matrix (NGTDM) (5 features), Laplacian of Gaussian (LOG) (180 features), and three-layer filtering wavelet (144 features) features (Appendix A).

Four readers with different levels of training performed manual segmentations on Neuroimaging Informatics Technology Initiative (NIFTI) format images and included a data scientist (BY) with no formal medical experience, a medical student (LS), a radiology trainee (MH) with 5 years of clinical radiology experience, and a specialty-trained thoracic radiologist (SK) with 18 years of experience. The data scientist (BY) used the snake feature of ITkSnap region growing tool, while he manually selected the region of tumors in the CT images, adjusted the contrast, set initial bubbles, controlled them to grow to a substantial size, and manually with a brush tool cleaned the areas that were not in the boundaries or exceeded them. The reader with the most experience (SK) was defined as the reference standard (RS) used for benchmarking. Prior to performing segmentations, each reader performed a NSCLC tumor segmentation in a training set of 10 cases from a different source (institution PACS system) supervised by the specialty-trained radiologist (SK) and received feedback on segmentation methods. After completing the training set, each observer completed segmentations of tumors for the complete data set of CT exams. The tumors were labeled in 3D on standard lung windows using ITkSnap (version 3.6.0) [24] by each reader. Segmentations were only performed once per patient per reader, taking breaks between segmentations at the discretion of the reader. A total of 429 radiomic features were extracted within the tumor volume of each image using the Pyradiomics library (v2.2.0) and analyzed using low-rank representations of radiomics using principal component analysis and selecting the first principal component (PC) corresponding to the maximum variance in the radiomics. The radiomic analyses were carried out in Python programing language (3.6.8), while the survival analyses were conducted in R programming software (4.0.1). Correlation between the extracted features and agreement between 3D segmentations were analyzed using a Pearson correlation coefficient and Sørenson-Dice coefficient [25], respectively. Dice coefficient measures variabilities of the segmented regions, and low-rank correlation shows its corresponding effect on radiomics by calculating correlation for direction of the maximum variances. In other words, correlation among three first PCs represent the correlation of the entire radiomics (all 429 radiomics). Appendix B provides additional information regarding principal component analysis (PCA). The proposed approach involves using machine learning to reduce the radiomic dimensionality and predict survival using PCA and Cox regression models, which increases the importance of applying unsupervised and supervised models’ integration.

Cox regression modeling was performed for each dataset, incorporating radiomic phenotypes, and clinical and demographic data (i.e., sex, stage status, and histology). Kaplan-Meier curves of overall survival were generated for each dataset to determine if contributing radiomic signatures were able to stratify high- and low-risk patients.

## 3. Results

### 3.1. Patient Population

A total of 89 patients were in the NSCLC- Radiomics-Genomics-Lung3 dataset, 3 of whom did not have available data and were excluded from the study. There were 42 patients with adenocarcinoma, 32 patients with squamous cell carcinoma, and 12 patients with another type of NSCLC. Thirty-nine patients had stage I disease, 26 patients had stage II disease, 10 patients had stage III disease, and 11 patients had an unknown stage. Of the NSCLC-Radiogenomics data in the NIH-TCIA dataset, 4 patients were excluded from the study for a total of 207 patients included. Of the included tumors in the Harvard dataset, all were solid, and of the included tumors in the Stanford dataset, 134 were solid, 68 were subsolid, and 5 were unknown.

The total number of patients included in the study is described in Figure 2. Clinical information and demographics of patients are provided in Table 1 and Table 2.

### 3.2. Analysis of Interobserver Variability on Radiomic Feature Extraction

From the 429 radiomic features initially extracted from the tumors on CT images, the feature-level was reduced to 3 radiomic signatures (three first PCs) for all the segmenters (Figure 1). The correlation coefficient among the low rank radiomic signatures showed significant correlation among the segmenters with a correlation of greater than 0.7 for all the cases (Table 3).

*Corr* coefficients using the first principal component between BY-SK (RS), LS-SK (RS), and MH-SK (RS) were 0.92, 0.94, and 0.95 (all having *p*-value < 0.005) for NSCLC-Radiomics-Genomics, and were 0.93, 0.72, and 0.87 (all having *p*-value < 0.005) for NSCLC-Radiogenomics, respectively, all indicating a strong correlation. The comparison of three significant radiomic descriptors corresponding with each group of segmentations showed 88.9% and 92.7% correlation of radiomics of each set with RS. Principal component analysis of the first three principal components demonstrates that, in some cases, there is a large standard deviation (STD), but the medians of the principal component analyses for the extracted features are similar and still have good correlation (Figure 3).

The Dice coefficients for the 3D masks for Harvard NSCLC-Radiomics-Genomics and Stanford NSCLC-Radiogenomics for each segmenter (Table 3) was 0.894 (STD: ±0.25) −0.71 (STD: ±0.28) for the image scientist (BY)—Reference Standard (SK), 0.82 (STD: ±0.14) −0.80 (STD: ±0.27) between the medical student (LS)—Reference Standard (SK), and 0.839 (STD: ±0.20) −0.83 (STD: ±0.23) between the radiology trainee (MH)—Reference Standard (SK), respectively. Although the SD coefficients indicate a moderately high spatial agreement of the segmentations, there was some variability between segmentations for BY-SK (RS), LS-SK(RS), and MH-SK (RS) (Figure 4). Precision of the analyses for all segmenters for both NSCLC datasets showed relatively similar precision in segmenting the tumors, where BY-SK(RS) in Harvard and LS-SK(RS) in Stanford datasets have the highest precision yielded to 81.8% (±21.8%) and 84.2% (±31.5%), respectively. MH-SK(RS) and BY-SK(RS) showed the highest recall with 88.7% (±18.9%) and 87.3% (±25.2%), respectively. This pattern showed consistency with the minimum volume difference for MH-SK(RS), 0.6(±1.9), in Harvard dataset, and BY-SK(RS), 0.3(±0.8), and LS-SK(RS), 0.3(±1.2), shared minimum volume difference in Stanford (See Table 3). We conducted in-depth correlation analysis for individual radiomics and showed the results based on radiomics’ categories (Appendix A). Moreover, we presented some radiomics that showed lesser stability among the segmenters in this study (Appendix A).

Cox regression modeling of overall survival for the NSCLC-Radiomics-Genomics-Lung3 (Harvard) and NSCLC-Radiogenomics (Stanford) datasets yielded a c-statistic of 0.64 (95% CI) and 0.6 (95% CI), respectively, for the model including only the clinical (sex, smoking status, and histology) and demographic covariates, which increased when adding radiomic signatures, having of c-statistic of 0.7 (95% CI) and 0.69 (95% CI), respectively. Adding clinical and demographic data to this model yielded an increase in c-statistic, although with slightly increased variability: 0.05–0.02 and 0.01–0.02 for NSCLC-Radiomics-Genomics-Lung3 and NSCLC-Radiogenomic datasets, respectively (Table 4).

Additional Cox regression analysis data are presented in the Appendix A. Kaplan-Meier curves of survival prediction for each dataset showed significant discrimination between high- and low-risk patients using extracted radiomic signatures (*p* < 0.01) and are presented in Figure 5. Median risk score was used as a distinguishing criterion for signifying high- and low-risk groups. The hazard ratio for each covariate in the maximal model is fully reported in the Appendix A.

## 4. Discussion

CT imaging is the workhorse of oncology staging and treatment response assessment. However, we now know that conventional imaging has imbedded “radiomic” features that are not appreciable by the eye but contain information on tumor heterogeneity that are reflections of the underlying tumor structure and can be harnessed to generate prognostic and predictive biomarkers. In addition, the morphologic qualitative descriptors used in conventional reporting of radiologic assessments of tumors on CT, such as “spiculated”, “heterogeneous”, and “necrotic”, while clinically useful, are subject to inter and intraobserver variability [10] due to their subjective nature; radiomic signatures may allow for more quantitative and precise measure of tumor description, potentially enhancing the clinical value of these interpretations.

In addition to providing a more quantitative approach to conventional morphologic descriptors, radiomics offers the potential to reveal aspects of tumor phenotype not discernable by the human eye, providing another layer of valuable information that can be extracted from conventional imaging for clinical management. Several studies have described the significance of these additional imaging features and radiomics in cancer imaging [26,27,28,29,30,31,32,33,34,35,36,37] and have hypothesized that tumor genetic and cellular characteristics and phenotypes can be represented with medical imaging [38,39,40]. For example, studies by Ganeshan et al. [41,42,43] reported an association of extracted NSCLC CT tumor features with patient survival, tumor stage, metabolism, angiogenesis, and hypoxia. The importance of imaging in treatment planning and outcomes was demonstrated by El Naqa et al. [44] for head and neck and cervical cancers, and Vaidya et al. [45] for lung cancer. Huang et al. [4] concluded that EGFR mutation status can be determined using quantitative imaging from extracted tumor phenotypes in NSCLC. Similarly, Bardia et al. [46] found that combining radiomic phenotypes, clinical variables, and circulating tumor DNA (ctDNA), enhanced prediction of EGFR-targeted therapy outcomes for NSCLC.

However, while the use of extracted radiomic features from conventional imaging poses exciting possibilities for precision medicine, there are challenges to clinical translation that must be overcome before the use of these novel techniques can become a reality in routine practice. There is variability introduced in the acquisition of imaging, for example the use of different imaging protocols, reconstruction algorithms, and scanner types. In addition, variability is introduced through choice of imaging processing techniques, such as choice of segmentation and feature extraction software, and degree of skill of the reader performing 3D segmentation. Variability is a particular concern with manual segmentations [47], and several studies have reported significant inter-clinician variation in contouring of tumors in radiation treatment planning, including head and neck, lung, prostate, and esophageal cancers [48,49,50,51,52]. In this study, we did find some variability between segmentations performed by the data scientist (BY), the medical student (LS), the radiology trainee (MH), and the most experienced reader, reference standard (SK). However, the SD coefficients suggest an overall moderate to high degree of spatial agreement of the segmentations and good overlap of tumor segmentations between readers.

Interobserver variability between readers in this study may have been introduced by several factors. One factor is differentiating between the boundaries of tumor and adjacent post-obstructive atelectasis [53,54] or pneumonia, a known problem with tumor delineation. In non-contrast CT examinations, it may also be difficult to delineate tumor and adjacent vascular structures that course in and adjacent to lung cancer, especially if the tumor abuts the hilum or mediastinum. Some lung cancers also demonstrated both a solid and a ground glass component, which can introduce variability in the choice of where to draw the boundary around faint ground glass components. Huang et al. [4] discovered that trained radiologists tended to focus on the solid component of a tumor as opposed to the ground glass component, whereas junior radiologists tended to include more of the ground glass component in their segmentations. The inclusion of more ground glass component would increase overall tumor volume and impact the spectrum of radiomic features extracted, thus a risk factor for variation. Window width and level settings on CT may also influence segmentations and gross tumor volumes [54,55,56,57]. ITkSnap software allows the reader to choose the window width and level settings in addition to an automatic window width/level selection. While some of our readers manually and arbitrarily adjusted the window width/level based on preference and ability to differentiate tumor from adjacent structures, other readers chose the automated window width/level setting chosen by the software.

Radiomic features used in this study follow imaging features defined by the Imaging Biomarker Standardization Initiative (IBSI). However, differences in CT exam parameters may also introduce segmentation variability between readers. This is particularly true with certain texture features such as coarseness and contrast, which tend to be the least reproducible. First order features, particularly entropy, are found to be the most reproducible [18]. Leijenaar et al. [58] found that radiomic features with high test-retest repeatability suffered less from interobserver differences. A few studies have confirmed that tube current (mAs) or tube voltage (kVp) had no influence on feature reproducibility [59,60]. Varying slice thicknesses of CT scans can also introduce variability in the extracted features, with 1–2.5 mm being the recommended slice thickness when contouring tumors [17,61]. Our study used a publicly available online dataset with slice thickness varying from 1–5 mm (Appendix A). We conducted in-depth analyses on the effect of CT parameters on the outcome of the selected features using the proposed approach and their final survival outcomes (Appendix A). Our supplemental analyses testing the potential effects of CT parameters indicated that there was an overall similarity among segmentations between readers when considering contrast-enhancement, CT kernel, and slice thickness.

The degree of medical specialty training has been a concern for the introduction of variability in segmentations of tumors. Logue et al. [62] reported that radiologists tended to contour smaller gross tumor volumes compared to radiation oncologists in the segmentation of bladder cancers and concluded that a more correct anatomic gross tumor volume was provided by radiologists likely due to clinical practice differences, since radiation oncologists typically select more inclusive volumes around tumors in practice so as not to underestimate tumor extent radiation treatment planning [63]. Similar results were observed in NSCLC by Giraud et al. [64], who noted major discordances between radiation oncologists’ and radiologists’ tumor delineations, radiologists tending to delineate smaller volumes. In this same study, junior physicians included as readers tended to delineate smaller and more homogeneous volumes compared to senior physicians regardless of their specialty. Van de Steene et al. [63] looked at specialty dependence between junior and senior radiation oncologists, one pulmonologist, and one radiologist, on contouring lung cancer gross tumor volumes and noticed that the radiologist ended up with the smallest tumor volume. They also noted good agreement between the senior radiation oncologist and radiologist. Haga et al. [65] concluded that NSCLC tumor volumes should be contoured by a specialist, such as a radiation oncologist, in order to decrease tumor delineation uncertainty and overestimation of prognostic power in radiomic feature analysis. In this study we compared tumor segmentations between level of training (i.e., medical student, radiology trainee, and radiology attending), and specialty type (i.e., data scientist). Interestingly, the 3D masks in the Harvard Dataset for BY-SK (RS) had an overall higher correlation compared to the masks for MH-SK (RS) and LS-SK(RS) in the segmentation analysis. However, the 3D masks in the Stanford dataset for MH-SK (RS) had an overall higher correlation compared to the masks for BY-SK (RS) and LS-SK (RS). The Pearson correlation coefficients, comparing three significant radiomic phenotypes for PCA, were all relatively equal amongst segmenters in the Harvard dataset, although the correlation coefficients were slightly more variable in the Stanford dataset. Overall, these differences are small and can probably be overlooked given overall high correlation of segmentations amongst all segmenters in the principal component analysis. It should be noted, however, that all readers in this study participated in a training set of cases supervised by the reference standard (SK) to ensure a standard approach to contouring.

Our study had several limitations. The CT scans in the dataset had varying slice thicknesses, ranging from 1–5 mm, which is known to introduce some variability as described above. Additionally, while all the readers used ITKSnap software for segmentation, there was some variability in methods of tumor contouring, such as choice of purely manual or semi-automated tools and the exact window and level used to perform the contouring. However, while there was interobserver variability in contouring, the extracted radiomic features of both the medical student, radiology trainee, and data scientist were overall well correlated with the experienced reader (RS). Another limitation is that the readers were all trained by the expert reader; however, the number of training cases was small and consisted of feedback of the segmentations. Additionally, the training cases were from a different source than the databases that were used for analysis. Despite the limitations, overall correlation of extracted features between readers supports the inclusion of readers of various levels of training in performing segmentations for NSCLC.

Future research would include testing interobserver variability based on level and type of experience against other publicly and readily available datasets and testing intraobserver variability. Other future directions should include determining how factors such as slice thickness, pixel spacing, window width/level, contrast enhancement, and pre- and post-processing of CT imaging affect interobserver variability between readers of different experience.

## 5. Conclusions

Although there is some variability in tumor contouring for imaging segmentations between readers, the extracted radiomic features were overall well correlated in observers. Therefore, level of training and clinical experience of the reader may not have a substantial impact on extracted radiomic features of NSCLC on CT, noting that all readers did have a supervised training set prior to contouring cases. Having more readers to perform tumor segmentations may accelerate the development of radiomic signatures in NSCLC that can provide added value to cancer management and precision medicine. This study shows that a greater degree of inclusion of personnel is allowable to perform these tumor segmentations.

## Figures and Tables

**Figure 1 cancers-13-05985-f001:**
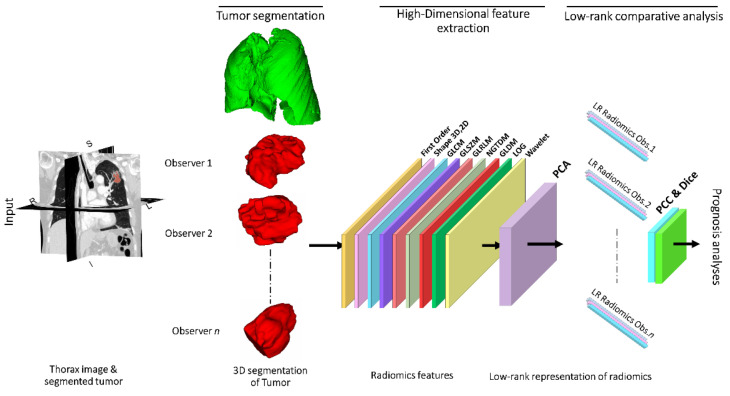
Workflow of the approach. The NSCLC tumor is segmented from the original CT images by four segmenters (*n* = 4) with different backgrounds, yielding radiomics features and tumor masks as inputs. Next, PCA categorizes features based on their maximum variance in radiomics. For every group, three principal components of feature sets are selected and used for correlative analysis and prediction of survival.

**Figure 2 cancers-13-05985-f002:**
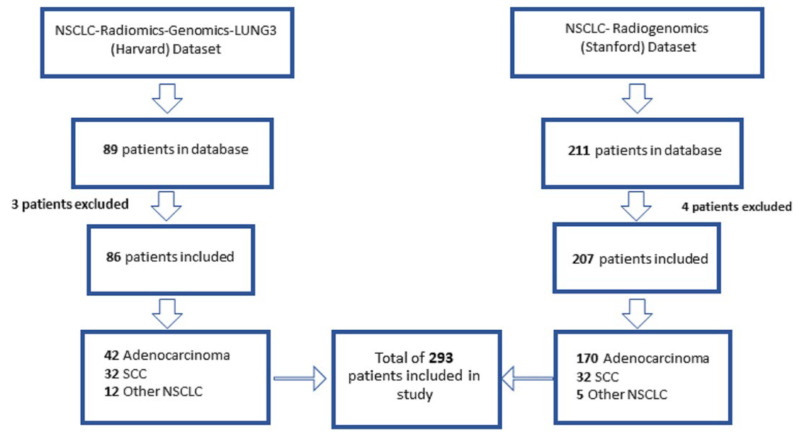
Number of patients included in study. Two publicly available datasets were analyzed in the study, the NSCLC-Radiomics-Genomics-Lung3 (Harvard) dataset and the NSCLC-Radiogenomics (Stanford dataset). Eighty-nine patients and 211 patients are part of the Harvard and Stanford datasets, respectively. A total of 3 patients were excluded from the Harvard dataset and 4 patients were excluded from the Stanford dataset due to lack of available data. Tumor types consisted of adenocarcinoma (Adeno), squamous cell carcinoma (SCC), and other types of NSCLC. A total of 293 patients were segmented as part of the study.

**Figure 3 cancers-13-05985-f003:**
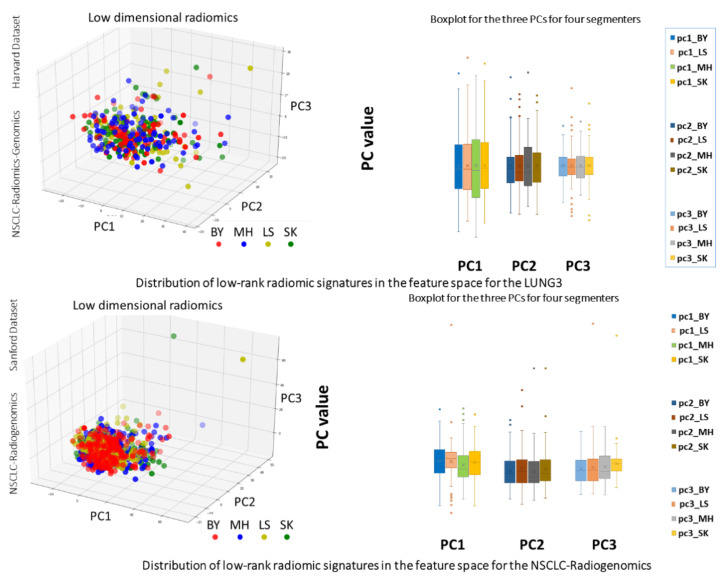
Two visual comparisons of low-rank radiomics representation with their boxplots relation for labels provided by BY, LS, MH, and SK for two different NSCLC Radiogenomics datasets.

**Figure 4 cancers-13-05985-f004:**
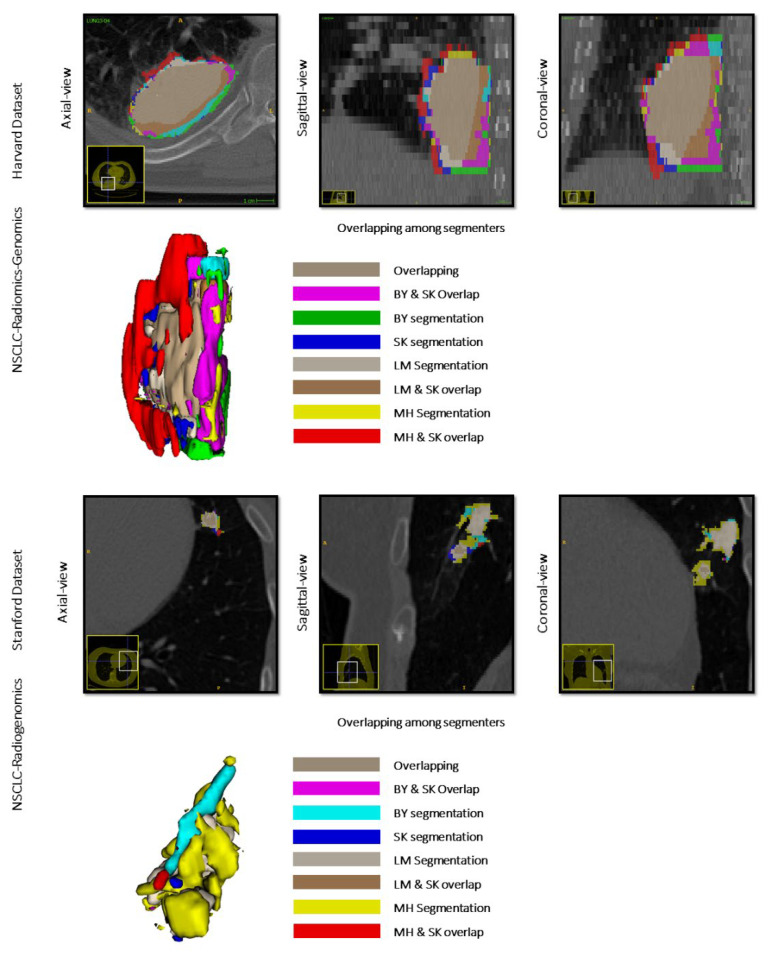
3D tumor volume. 3D tumor volumes for four segmentation cases and two different NSCLC Radiogenomics datasets.

**Figure 5 cancers-13-05985-f005:**
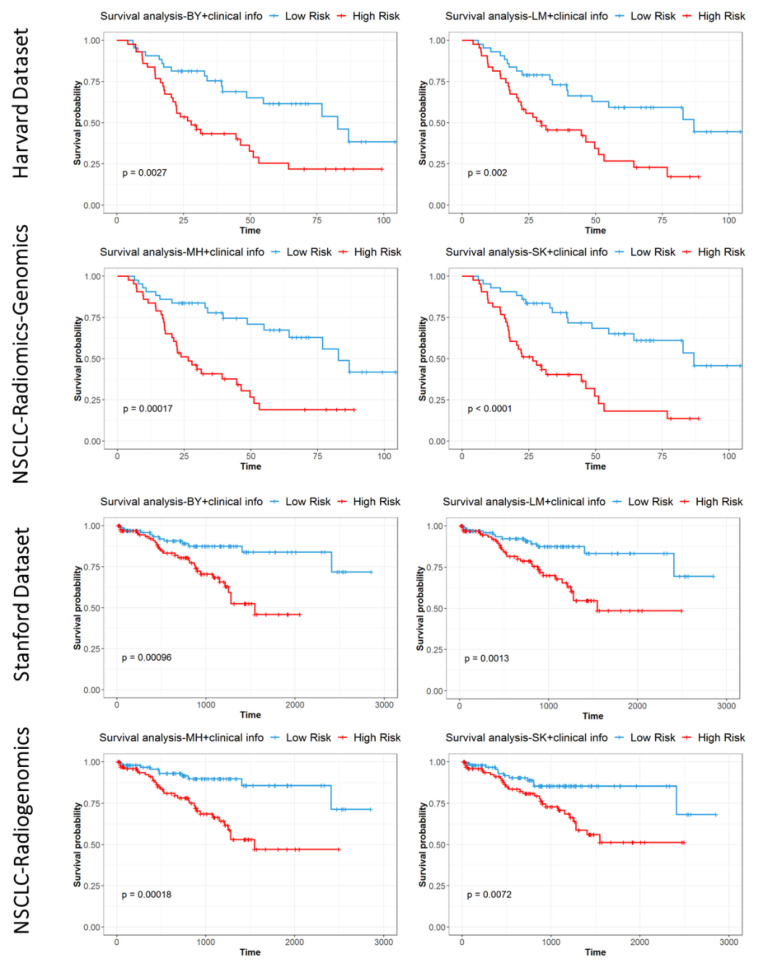
Kaplan-Meier curves for multivariate models of overall survival using low-rank radiomics show significant differences between high- and low-risk patients for each segmenter and NSCLC dataset using median risk score in the model.

**Table 1 cancers-13-05985-t001:** Clinical and demographic data including gender, type of NSCLC, and stage of cancer for collected patients in NSCLC-Radiomics-Genomics (Harvard) lung dataset is presented.

NSCLC-Radiomics-Genomics		
Gender	Male Female	61 (68.5%) 28 (31.5%)
Clinical combined stage curated	Stage I Stage II Stage III Unknown	39 (43.8%) 25 (28.1%) 12 (13.5%) 11 (12.4%)
Non-small cell lung cancer (NSCLC)	Adenocarcinoma, Squamous cell carcinoma Other or unknown	42 (47.2%) 33 (37.1%) 12 (13.5%)
Event	Recurrence or death	46 (51.7%)

**Table 2 cancers-13-05985-t002:** Clinical and demographic data including age, race, type of NSCLC, EGFR, and KRAS receptor status, and smoking status for collected patients NSCLC-Radiogenomics (Stanford) is presented.

NSCLC-Radiogenomics		
Age	Median (±IQR)	69 (43,87)
Gender	Male Female	133 (64.2%) 74 (35.8%)
Race	Caucasian Asian Hispanic/Latino African-American Native Hawaiian/Pacific Islander Unknown	120 (57.4%) 24 (11.8%) 5 (2.4%) 6 (2.9%) 3 (1.5%) 48(23.2)
Smoking Status	Non-smoking Smoking Former smoking	47 (22.7%) 34 (16.4%) 126 (60.9%)
EGFR-Mutation Status	Wildtype Mutant Unknown	128 (61.8%) 42 (20.2%) 37 (17.8%)
KRAS Mutation Status	Wildtype Mutant Unknown	130 (62.8%) 38 (18.3%) 39 (18.8%)
Histology	Adenocarcinoma Squamous cell carcinoma NSCLC NOS (not otherwise specified)	170 (82.1%) 32 (15.5%) 5 (2.4%)
Solid-Subsolid (Morphology)	Solid Subsolid Unknown	134 (64.7%) 68 (32.8%) 5 (2.4%)
Event	Recurrence or death	41(21.1%)

**Table 3 cancers-13-05985-t003:** Similarity of the radiomic signatures using multiple scoring methods among different segmenters are presented.

NSCLC Dataset	Similarity among Segmenters						
	Segmenters ID	Correlation Score	Dice Score	Precision(%)	Recall (%)	Boundary Distance	Volume Difference
**LUNG3** **NSCLC-Radiomics-Genomics** **Harvard Dataset**	BY	0.92	0.89 (±0.25)	81.8 (±21.8)	86.1 (±24.5)	1.2 (±2.7)	1.1 (±0.5)
	LS	0.94	0.82 (±0.14)	81.2 (±2.7)	69.6 (±24.5)	6.5 (±26.4)	2.3 (±21.1)
	MH	0.95	0.84 (±0.20)	72.3 (±22.4)	88.7 (±18.9)	4.2 (±15.1)	0.6 (±1.9)
**NSCLC-Radiogenomics** **Stanford Dataset**	BY	0.93	0.69 (±0.28)	77.8 (±25.1)	87.3 (±25.2)	2.92 (±10.7)	0.3 (±0.8)
	LS	0.72	0.80 (±0.27)	84.2 (±31.5)	47.8 (±29.9)	16.6 (±52.6)	0.3 (±1.2)
	MH	0.87	0.83 (±0.23)	80 (±24.3)	77.1 (±24.7)	6.2 (±26.1)	1.4 (±16.9)

**Table 4 cancers-13-05985-t004:** Overall survival, Cox regression. Using the low-rank representation of the radiomic signatures survival prediction is measured for each segmenter.

Prediction Survival									
NSCLC Datasets	Modeling Covariates	BY		LS		MH		SK-RS	
		c-Statistic (95% CI)	*p* Versus Null ^1^	c-Statistic (95% CI)	*p* Versus Null ^1^	c-Statistic (95% CI)	*p* Versus Null ^1^	c-Statistic(95% CI)	*p* Versus Null ^1^
**LUNG3** **NSCLC-Radiomics-Genomics** **Harvard Dataset**	clinical and demographic ^2^							0.64	0.2
	Three PC radiomic signatures	0.6	0.5	0.62	0.08	0.59	0.2	0.65	0.03
	Radiomic signatures, clinical and demographic	0.65	0.3	0.68	0.04	0.66	0.2	0.7	0.03
**NSCLC-Radiogenomics** **Stanford Dataset**	clinical and demographic ^3^							0.6	0.007
	Three PC radiomic signatures	0.65	0.001	0.64	0.04	0.67	0.003	0.65	0.003
	Radiomic signatures, clinical and demographic	0.71	<0.005	0.68	0.003	0.71	<0.005	0.69	<0.005

CI: confidence interval. ^1^
*p*-value by likelihood ratio test versus the hypothesis that the model is no better than the null model. ^2^ Clinical and demographic covariates for LUNG3-NSCLC-Radiomics-Genomics Harvard Dataset: sex, stage status, and histology. ^3^ Clinical and demographic covariates for NSCLC-Radiogenomics Stanford Dataset: sex, morphological status, and histology.

## Data Availability

Information on the publicly available datasets used in this study [19,20,21].

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
