# Peer review of "Impact of Interobserver Variability in Manual Segmentation of Non-Small Cell Lung Cancer (NSCLC) Applying Low-Rank Radiomic Representation on Computed Tomography"

_cancers, 2021, doi:10.3390/cancers13235985_

Round 1
Reviewer 1 Report
Thank you for this interesting work.
Some comments:
- how was the patient example decided, it says 89? have to be clarified.
- It is also important to know if the images choosed were at first presentation of the patient in the diagnostic process? Have any of the patients included undergone any therapy?
- was the segmentation done repeatedly or just a single event?
- some thoughts about how the authors want to go further in implemanting the findings
- how is this work connected to "machine learning", would be interesting to hear some comments as this topic also is largerly discussed regarding radiology
Reviewer 2 Report
This is a study looking to investigate interobserver variability of manual segmentations in non-small cell lung cancer on CT imaging. A radiologist along with three trainees segmented 295 patients and segmentation metrics were analyzed. Radiomic features were extracted and PCA performed. A Cox model and Kaplan Meier analysis was also performed to predict overall survival. The variability in segmenting lung tumors is well known and although this study compares different expertise of users, I have concerns regarding the novelty and overall conclusions. The methodology has several details missing to accurately assess what was performed and there are several limitations as described below.
Major Comments:
- Introduction: The description of the radiomic features used in this study should be left for the methods. The introduction would benefit from a more thorough description of the unmet need and advancement in knowledge this paper is going to address. Specially given that most studies suggest the use of semi-automatic segmentation tools, which are known to be more reproducible than manual segmentation.
- Methods – Patient Population: It would be helpful to describe the types of tumors in each dataset. For example, were they all solid tumors or did these datasets also include ground-glass tumors?
- Methods – Segmentation: The most experienced user was used as the reference standard for benchmarking. However, it is well known that extensive inter- and intra-observer variability can exist among radiologists. Similarly, each user was trained by this experienced radiologist. This could substantially bias and impact the results of this study. The results therefore do not give a true sense of the inter-observer variability you would expect to see across random observers.
- Methods – Segmentation Variability: The authors have only performed a segmentation comparison using the Dice coefficient. Dice is known to have limitations and a more well-rounded set of segmentation metrics should be analyzed for a true representation of the variability between observers (e.g., boundary distance, volume difference, precision, recall).
- Methods – Radiomic Features: For reproducibility, all parameters and details of the radiomic feature extraction need to be provided. For example:
- What version of Pyradiomics was used?
- Was any pre-processing done on the images (e.g., voxel resampling)?
- What were the feature extraction parameters (e.g., bin width)?
- Methods – Correlation: Can the authors please justify the use of PCA for the correlation of features. A limitation of the current study is it does not provide any insight into which specific types of radiomics features may be more or less stable. Do the authors have any insight to the specific features that were correlated? Similarly, an ICC analysis could also be performed to assess the consistency of features from different segmentations.
- Methods – Cox: Additional information is needed on the Cox modelling.
- What clinical and demographic data was included in the Cox models?
- What software was used to perform the modelling?
- Figure 4: How were the cut-off points to separate high and low-risk groups for the Kaplan-Meier curves determined (e.g., median risk score)? And was this value the same for all plots? For a true comparison across observers, the same cut-off value should be used.
Minor Comments:
- Introduction: The description of the radiomic features is not completely accurate. For example, the authors state there are nine families of features. However, the Laplacian of Gaussian and wavelet features are not a feature class, but rather a filter applied to the images. Therefore, the previous family of features are also calculated on these filtered images. This needs to be clear when describing the features.
- Figure 1: It is unclear the purpose of the rendering below the CT image, since the tumor segmentation is not visible.
- Methods – Patient Population: This is a bit repetitive when talking about the NSCLC Radiogenomics cohort, as NIH and TCIA is repeated twice.
- Result – Page 8, lines 208-212: Consider simplifying this paragraph to avoid repetition as the same sentence is repeated twice.
- Introduction and Discussion: The authors have briefly mentioned previous studies looking at segmentation variability. However, a more thorough description of the previous literature in lung cancer is warranted to put these results in the context of previous studies.
Reviewer 3 Report
the manuscript is very well prepared and exhaustive in my opinion.
the topics (radiomic and inter observers variability) are of great interest in lung cancer oncology despite not still applicable in routine practice.
I very much appreciated the materials and method section (retrospective review within a large series of cases), coming from a worldwide renown institution (Harvard and Stanford databases). This should be considered an adjunctive element in evaluation (data from histology, surgery, etc are likely to be of high quality)
the results - despite complexity of this field - are clearly exposed and available also for non-radiologists specialists (oncologists, surgeons)
i do not have any specific recommendations to ask/add to the manuscript that in my opinion deserves outright publication
my only comment is on radiomic itself: this is a new tool still under assessment and still not part of clinical practice. The main problem is not only inter observer variability; the relevant critical issue is the ability of this modern technology to adequately anticipate tumor histogy subtypes, mutational status, therapeutic consequences, more in general the capability to reproduce before any treatment start a precise staging and biology of the tumor. This is greatly promising but still under construction. I congratulate the Authors for this study because inter-observer differences is for sure part of the game but I am not convinced is the more important part. For instances: segment origin of a lung cancer is important but less crucial than immunological status or degree of infiltrative cancer in GGO.
Everything should be studied and improved: still the “package” of information we expect from radiomic when included in lung cancer initial assessment is wide and includes most of all the anticipation of biology behavior.
Thanks for letting me review this interesting manuscript
Reviewer 4 Report
The objective set by the authors for this paper is an important one - Investigate the segmentation variability across several experts that have varying degree of experience. Thorough evaluation of the segmentation variability and its effects on the radiomics pipeline is important and needed.
Well-written and well organized paper. Following are my comments. Where L stands for the Line.
L30. Why does the abstract say three readers? Weren't four readers involved?
L40. Very strong conclusions, suggest toning it down. Especially given the fact that authors have investigated only one use case of NSCLC on only one imaging protocol of CT. Results could have been different if the study was conducted for Glioblastoma or pancreatic cancer on MRI or other imaging modalities. This study is specific to NSCLC on CT only.
Lack of consistency between Table 1 and Table 2- I understand that these publicly available datasets might have different information available from them. But the authors should make an effort to combine these tables and present common information.
Section 3.1 : Suggest creating a flowchart figure that explains the data involved in this study-- The flowchart can detail the inclusion/exclusion criteria with the associated 'n' at each step. In it's current format, it is a little difficult to follow the several cohorts from which the data was curated and used for this study.
Section 3.2: One of the main issues I have is the decision to do a PCA and reducing it to a radiomic signature. If the main objective of the paper was to understand the direct relationship of segmentation variability with radiomic features extracted-- then why introduce a layer of abstraction with PCA? Why didn't the authors investigate the individual correlation of each of the 429 radiomic features? The authors could have also identified feature family specific trends. Maybe the GLCM features were more resilient to segmentation compared to LoG or wavelet features. Why reduce everything to an abstracted radiomic signature?
L.127 - Further details are required regarding this setup. Did the observers take breaks? Was the task completed during one sitting? Have the authors looked into intra-reader variability as well?
L.211 - is it a typo? Seems repeated
Round 2
Reviewer 2 Report
Thank you to the authors for their comprehensive response to the previous reviewer comments. The additional information that was added to the methodology has improved reproducibility and strengthened the results and conclusions. The authors have also addressed some of the previous limitations in the discussion. However, there were still some comments that were not thoroughly addressed, and new concerns raised based on the authors responses.
- Methods – page 4, lines 136-142: The description of the number of radiomic features used is suitable for the methods. However, the authors have also moved a summary of the issues with reproducibility. This section is not suitable for the methods and should be included in the introduction or discussion.
- Methods – page 4, lines 153-154: The authors have made an important point in their response to my comment that the semi-automatic approach in ITK-Snap is similar to a manual segmentation due to the significant user interaction. However, it was not clear or mentioned anywhere in the methods that users used this technique. Based on the title and purpose in the introduction, I assumed it was fully manual segmentation.
- Since ITK-Snap allows for solely manual segmentation or active-contours, the precise methods used by the readers should be included in the methodology. For example, could they choose either one or were they told to use active contours?
- Also, using the active-contour option isn’t truly a manual segmentation, as this could decrease variability given the semi-automated nature of the tool. These details need to be well described and clarified throughout the manuscript. Currently this is a major limitation that could bias the results, while contradicting the authors use of manual segmentations throughout.
- Table 1: The authors mentioned that the tumors were predominantly solid nodules. Can the percentage of patients with solid or ground-glass nodules be added to Table 1?
- Methods – page 4, lines 156-157: Thank you for providing more details on pre-processing and version numbers. However, some parameters of feature extraction are still missing.
- For example, what were the parameters used (e.g., bin width or bin numbers). There also needs to be details on the size of the LOG filters that were applied. For ease of reproducibility the Pyradiomics parameter file could be included in the supplementary material.
- Similarly, the categorization of features by Pyradiomics separates the filters (https://pyradiomics.readthedocs.io/en/latest/features.html). It needs to be clear which of these feature types were calculated on the wavelet and LOG images.
- Figure 2: The overall quality of this figure is low with the different colours and text overflowing the boxes. Consider simplifying to improve readability and interpretation.
- Figure 3: The font in this figure is illegible. Considers increasing all font sizes for readability.
- Figure 4: The authors clarified that the median risk score was used in their response document, however this detail was not added to the manuscript. For reproducibility, please state that the median risk score was used and what the exact value was for potential validation in future studies.
Reviewer 4 Report
Thank you for your response. All my concerns have been addressed.
Round 3
Reviewer 2 Report
Thank you to the authors for their response to my previous comments. Although some of the comments have been addressed, there are still comments which were not sufficiently addressed. Also, these changes were not tracked in the updated manuscript, therefore making it hard to determine what changes were made since the last revision.
- For example, the authors have clarified the median risk score and I can see this has been added on page 10 (not tracked). However, the exact risk score that was used was not provided.
- The precise risk score value that was used to separate into low and high-risk groups needs to be stated for each plot for future studies to validate or reproduce this work.
- It is also unclear what the authors mean by “median survival prediction of 0.5 for 30-68 months.” There are six plots provided, therefore for which plot and which curve?
- Regarding using the term manual segmentation, I respectfully disagree with the authors as the link they provided to ITK-Snap specifically states, “automatic segmentation using region competition snakes.” In their response the authors state that users performed a “complete manual segmentations of the tumor” however all of those steps are semi-automatic and a fully manual approach would be with no assistance. I completely understand the amount of user interaction that is required. However, with the region growing there is still some aspect of automatic segmentation (snake evolution to define the boundary of the segmentation) which can decrease the variability between users (the users are not defining this boundary). Overall, what the authors have described is a semi-automatic approach for segmentation. This is reasonable, however the terminology in the manuscript needs to reflect what was actually done. Also, the authors state they have modified the manuscript and clarified this; however, I do not see any changes that were made (e.g., in section 2.2).
- Thank you for providing the feature extraction parameters in the supplemental material. However what sigma levels were calculated (e.g., 1-10mm)? This needs to be stated and added to the table.
